# A Hepatogastrophrenic Trunk, Celiacomesenteric Trunk, and a Middle Mesenteric Artery in a 68-Year-Old White Male Donor

**DOI:** 10.3390/diagnostics12071597

**Published:** 2022-06-30

**Authors:** Ariana Sheridan, Elizabeth Reynolds, Elizabeth Maynes, Gary Wind, Maria Ximena Leighton, Guinevere Granite

**Affiliations:** 1Directorate for Education, Training, and Research, Walter Reed National Military Medical Center, Bethesda, MD 20814, USA; ariana.sheridan@gmail.com; 2F. Edward Hebert School of Medicine, Uniformed Services University of the Health Sciences, Bethesda, MD 20814, USA; elizabeth.reynolds@usuhs.edu; 3Department of Surgery, Uniformed Services University of the Health Sciences, Bethesda, MD 20814, USA; elizabeth.maynes@usuhs.edu (E.M.); gary.wind@usuhs.edu (G.W.); maria.leighton@usuhs.edu (M.X.L.)

**Keywords:** hepatogastrophrenic trunk, celiacomesenteric trunk, middle mesenteric artery, gastrointestinal vascular variations

## Abstract

A detailed understanding of the enteric vascular supply is of great importance for pre-operative planning. In the case of this 68-year-old white male donor, the following variations were observed: a hepatogastrophrenic trunk, a celiacomesenteric trunk, and a middle mesenteric artery. Literature review was conducted to understand the frequency and clinical significance of these variations.

## 1. Introduction

Generally, there are three unpaired branches arising from the abdominal aorta (AA): the celiac trunk (CT), the superior mesenteric artery (SMA), and the inferior mesenteric artery (IMA) [1,2]. Variations in this classic organization are of clinical importance when considering the blood supply to the enteric tract, planning for surgical and interventional radiology procedures, and more.

These variations also include the origin and branching pattern of the inferior phrenic arteries (IPAs). Generally, they arise from the AA or CT (18–30%) and contribute to the blood supply of the diaphragm, esophagus, stomach, liver, and adrenal glands [2,3,4,5,6,7]. The right and left IPAs may arise independently (62%) or from a common trunk (55%) [8]. This common trunk may give rise to additional arteries. For instance, hepatogastrophrenic trunks (HGPTs) have rarely been described in the literature and may have a variety of arrangements all stemming from embryologic aortic roots that have undergone anomalous coalescence or persistence [9]. 

The CT typically gives off the left gastric artery (LGA), common hepatic artery (CHA), and the splenic artery (SA), termed the hepatogastrosplenic (HGS) trifurcation per the Adachi classification system or the Tripus Halleri by Haller [10,11,12]. Among the many variations of the CT is the celiacomesenteric trunk (CMT): a common trunk of the CT and SMA. This variation is particularly rare, with an estimated prevalence of 0.54–3.4% [13]. Importantly, any vascular insult to this area would result in significant disruption of all supplied areas of the hepatobiliary and enteric system. 

The proper hepatic artery (PHA) bifurcates to form the left hepatic artery (LHA) and the right hepatic artery (RHA). Hepatic arteries can be replaced, which means that they arise from an anomalous vessel and supply a portion of the liver. Hepatic arteries can also be accessory vessels, which means they can arise from an anomalous vessel but supply a portion of the liver with another vessel [14]. LHA variations have been cited in multiple case reports and literature reviews [14,15].

The presence of a middle mesenteric artery (MMA) is a rare anomalous branch of the AA. It most commonly supplies the transverse colon, effectively replacing the middle colic artery (MCA) [16,17,18,19]. MMA incidence is estimated to be <0.1% [20]. This article explores these three distinct variations of the unpaired aortic branches: the HGPT, CMT, and MMA that all occurred in the same individual.

## 2. Case Description

The branching pattern variations observed in the 68-year-old white male donor (listed cause of death of chronic obstructive pulmonary disease (COPD) and pneumonia) were as follows. A common trunk for the right and left IPAs, an accessory LHA (ALHA), and the LGA originated directly from the AA as a hepatogastrophrenic trunk (HGPT) measuring 0.9 cm in width (Figure 1 and Figure 2). There was a common trunk for the CT and SMA known as a celiacomesenteric trunk (CMT) measuring 1.3 cm in width (Figure 2 and Figure 3). The CT (1.15 cm) had the following branches: the SA (1.1 cm) and the CHA (1.2 cm). The left gastro-omental (gastroepiploic) artery (LGOA) was a branch of the SA. The CHA branches were the right gastric artery (RGA) and the gastroduodenal artery (GDA). The CHA then became the PHA with the following branches: a common trunk of the cystic artery (CA) and the RHA, as well as the LHA (Figure 1, Figure 2 and Figure 3). The GDA branches were the superior pancreaticoduodenal arteries and the right gastro-omental (gastroepiploic) artery. This donor was one of fifty acquired from the Maryland State Anatomy Board for the Class of 2025 first year medical school anatomy courses.

There was also a middle mesenteric artery (MMA) (0.4 cm in width) that originated directly from the AA between the SMA (1.0 cm) and IMA (0.5 cm) (Figure 4). It supplied the transverse colon with the MCA and another branch that helped to create the marginal artery of Drummond (MAD) with the left colic artery (LCA). The IMA had a normal branching pattern with the LCA, two sigmoid arteries (SiA), and the superior rectal artery (SRA) (Figure 4). There were no SMA and IMA anastomoses present. The AA had a slight aneurysm just below the IMA, measuring 2.4 cm at the aneurysm site and 1.9 cm more proximally. The AA also demonstrated marked atherosclerosis.

## 3. Discussion

### 3.1. Celiacomesenteric Trunk (CMT)

The CT has an abundance of variations that were first classified by the Adachi Classification System [10,12]. This system outlined five variations, most common of which is the classically described HGS trunk, with a prevalence of 86% in their study population of 252 dissections of Japanese cadavers [12]. Trunk classifications 2 through 5 include: hepatosplenic (8%), gastrosplenic (3%), CMT (1.5%), hepatosplenomesenteric (1%), and hepatomesenteric (0.5%). Many additional variations have since been described. Gielecki et al. (2005) outlined a new classification system based on the number of CT branches numbered 1 (trifurcation) through 6 (hexafurcation), which were further subdivided based on the most common arterial divisions [12]. This study is not the only one to propose a more robust classification system. For example, Tang et al. (2019) also proposed a classification system based on angiographic studies [21]. They named five variations: hepato-gastro-mesenteric (type 1), hepato-spleno-mesenteric (type 2), gastro-spleno-mesenteric (type 3), hepato-gastro-mesenteric (type 4), and any other variation (type 5). Per this definition, the donor in this study had a type 5 CT with branches: SMA, SA and PHA, with a separate common trunk of the RIPA, LIPA, LHA and LGA.

Additionally, this donor had a common trunk for the CT and SMA, referred to as a CMT, which occurs in less than 1% of all anomalies involving the celiac axis [22]. This trunk also provided branches to the SA and PHA. The CMT has an incidence estimated to be less than 0.54–3.4% of the population [13]. This variation may remain asymptomatic as presumed in this patient, incidentally discovered via angiography, or involved in disease processes such as occlusive disease, aneurysms, or dissection. In one case report, thrombosis to this common trunk resulted in a widespread infarction [23].

### 3.2. Hepatogastrophrenic Trunk (HGPT)

The IPAs may arise independently from the AA or as CT divisions. The presence of a HGPT, as seen in this donor, has been cited in only a few case reports. Prevalence has been cited to be as low as 0.3% and as high as 1% [24]. According to Aslaner et al. 2017, when both IPAs originate from a common trunk, the most common location for the trunk is the AA [25].

### 3.3. Accessory Left Hepatic Artery (ALHA)

This donor had an ALHA arising from the HGPT. In general, hepatic artery variations are common. Abdullah et al. (2009) described 932 cases of liver transplants and HA variations were found in 31.9%: CHA anomalies, LHA and RHA anomalies to include both replaced and accessory arteries [26]. Specifically, literature review yielded multiple case reports of an ALHA. Hardy and Jones (1994) reported that the incidence of an aberrant LHA is 19% in 180 patients undergoing liver transplantation [27]. In 15% of the cases, the accessory artery arose from the LGA, and either the SA, GDA, or from the AA in 4% of the cases. For instance, Pai et al. (2008) described a case report of an ALHA arising from the CHA [28]. Noussios et al. (2017) conducted a literature review of anatomic variations of the LHA found on angiography and described a replaced LHA stemmed from the LGA in 3% of cases [29]. An ALHA was present in 3.2% of cases [29]. Another case report describes an ALHA arising from the CHA [28].

### 3.4. Middle Mesenteric Artery (MMA)

Lawdahl et al. (1987) first identified the term middle mesenteric artery (MMA), describing an artery arising from the AA anywhere between the SMA and IMA and supplying variable segments of the bowel [17]. In most individuals, the SMA gives rise to the MCA, which joins with the LCA (arising from the IMA) to anastomose and form the MAD. The MMA may also be an anomalous origin of the MCA [18]. This network is the major vascular supply to the transverse and descending colon. In 3–5% of individuals, the MCA is absent, but otherwise, variations in the MCA are not well described [20,30]. This donor had an independent branch off the aorta between the SMA and IMA, termed the MMA. The presence of an MMA is a rare anomalous branch of the AA, effectively replacing the MCA [16,17,18,19]. The incidence of this variation type is estimated to be <0.1% [20].

### 3.5. Embryonic Development

Embryologically, ventral and dorsal paired segmental arteries develop from the paired dorsal aorta at every vertebra. The vitelline arteries are initially paired, and LGA, CHA and SA arise from the 10th segmental vessel, and SMA from the 13th. The IMA arises from the 22nd vitelline artery. The 10th through 13th ventral segmental arteries provide the blood supply for the foregut and midgut, with an arterial anastomosis running parallel to the aorta connecting the roots during fetal development. Variations in celiac and mesenteric anatomy are thought to result from alterations in the regression of these segmental roots in combination with persistence of the longitudinal anastomotic artery. In regard to the presence of the MMA, this is thought to derive from embryologic failure of regression of the vitelline artery located between the 13th and 21st vitelline segments [31].

### 3.6. Clinical Significance

#### 3.6.1. CMT

This variation may remain asymptomatic as presumed in this patient, being incidentally discovered via angiography, or involved in disease processes such as occlusive disease, aneurysms, or dissection. In one case report, thrombosis to this common trunk resulted in a widespread infarction [23]. Celiac trunk stenosis or occlusion is reported to accompany an aberrant CMT. Visceral artery aneurysms are rare, and even rarer, are aneurysms associated with this CMT [32].

Specifically with the use of the Appleby or Modified Appleby procedure, originally designed for advanced gastric cancer, and also used in cases of pancreatic body and tail ductal adenocarcinomas with vascular invasion of the celiac axis, the value of knowing the variant anatomy is essential in determining the appropriate resection and reconstruction [33].

In the absence of anastomosis between the SMA and IMA, there is no redundancy in flow. In the case of the CMT, there is no redundancy between the CT and SMA. In these cases, proximal stenosis or occlusion would likely have severe ischemic consequences to the intestines, whether it be acute or chronic mesenteric ischemia. In cases with multiple variants without redundancy, the length of intraoperative clamping time may result in postoperative ischemia with downstream insufficient blood supply. Awareness of individual vascular anomalies may assist in avoiding postoperative gangrene and necrosis of the intestines.

#### 3.6.2. HGPT

The common trunk of right and left IPAs has particularly important clinical significance when considering treatment of hepatocellular carcinoma. The right IPA can provide parasitic supply to this tumor, particularly those at the dome of the liver and bare area, and in cases where resection is not feasible, IPA embolization is an alternative treatment modality [34]. For this reason, precise identification of the artery and its origin is critical.

#### 3.6.3. ALHA

Knowledge of ALHA variations is important when performing hepatobiliary surgery and preventing liver ischemia. In fact, one study suggests that up to 20% of ALHA originating from the LGA, supply at least two liver segments. The LGA, therefore, should be preserved when undergoing lymph node dissection in gastric cancer cases in order to protect liver vascularization [35]. The ALHA is also known to create difficulty during laparoscopic gastric banding operations due to its presence in the pars condensa of the lesser omentum. This procedure requires meticulous identification of arteries and consideration of placement of the band [36].

#### 3.6.4. MMA

While the presence of an MMA in and of itself is not of particular significance, the presence in relation to the absence of other vessels, and the territory which it supplies is important diagnostically and therapeutically. Additionally, there have been case reports of colonic hemorrhage from the MMA [19].

#### 3.6.5. Other

Persistent embryonic arteries and the final vessels that form may be predisposed to aneurysm. This is due to the increased likelihood of congenital defects in their arterial elastic and smooth muscle layers. This is well-studied in the case of a persistent sciatic artery and the frequency of aneurysms in this aberrant vessel. Ailawadi (2004) presents a cadaver with multiple aberrant vessels from embryologic error that may have been at increased risk for aneurysm formation [37]. Additionally, underlying congenital defects of these aberrant arteries may lead to aortic dissections, and a CMT may be a marker for increased risk for aortic dissection [38].

Knowledge of vascular anomalies is essential for navigation in many surgical and radiologic procedures. The importance of vascular imaging and identification of anomalies prior to surgical procedures cannot be minimized. Knowing the variations may influence the approach to treatment of many diseases, especially ones that involve catheterization of arteries supplying the small bowel, or methods of endovascular management. Surgical procedures such as lymphadenectomy surrounding the abdominal vasculature, aneurysm repair, aortic replacement, and chemoembolization are just a few of the operative procedures that may benefit from preoperative imaging. Preoperative diagnostic scans with CT angiography may provide better visualization of vascular anomalies in certain clinical cases and purport surgical decision making, such as endovascular to conventional surgery. Recognizing variant anatomy and anomalous vessels has important consequences both diagnostically and therapeutically.

## 4. Conclusions

The presence of multiple gastrointestinal vascular anatomical variations in a 68-year-old white male donor emphasizes the importance of anatomical study for future health care providers, preoperative diagnostic imaging before surgical intervention, and intraoperative awareness of the array of vascular anomalies that may be present. The frequency for these multiple variations to occur independently in one individual is estimated to be very low and has not increased despite improved imaging techniques. It is more likely that these variations are due to disruptions in the embryonic development of the enteric circulation, causing the rare combination of arterial variations presented in the current case. This case adds to the growing list of existing complex variations to the celiac axis and the importance of imaging prior to surgical intervention, particularly in abdominal approaches.

## Figures and Tables

**Figure 1 diagnostics-12-01597-f001:**
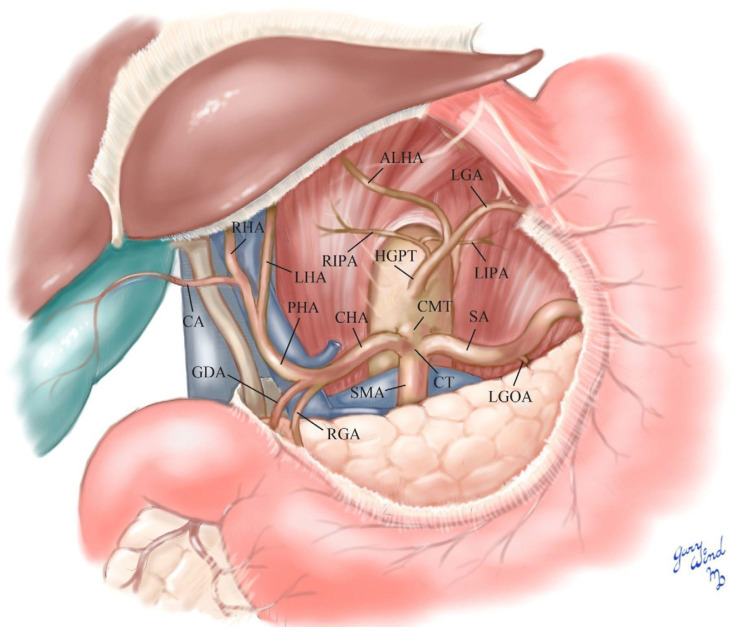
Schematic of donor abdominal vasculature.

**Figure 2 diagnostics-12-01597-f002:**
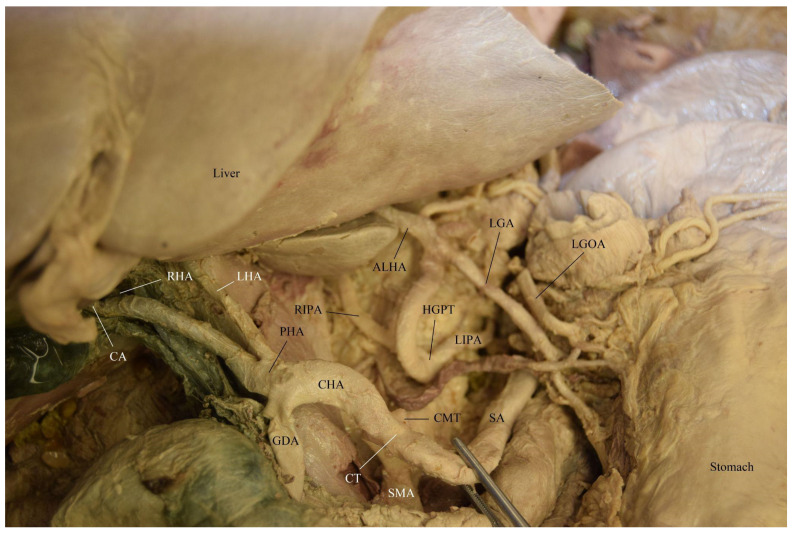
Arterial vasculature of the hepatogastrophrenic trunk (HGPT) and the celiac trunk (CT) of the celiacomesenteric trunk (CMT).

**Figure 3 diagnostics-12-01597-f003:**
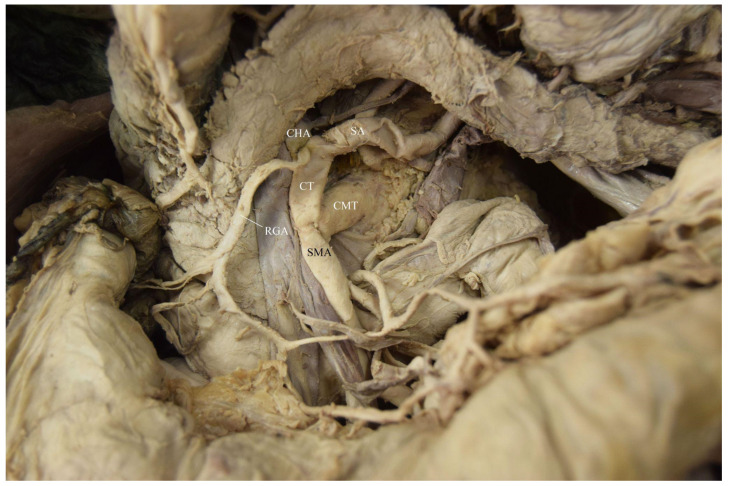
Arterial vasculature of the celiacomesenteric trunk (CMT).

**Figure 4 diagnostics-12-01597-f004:**
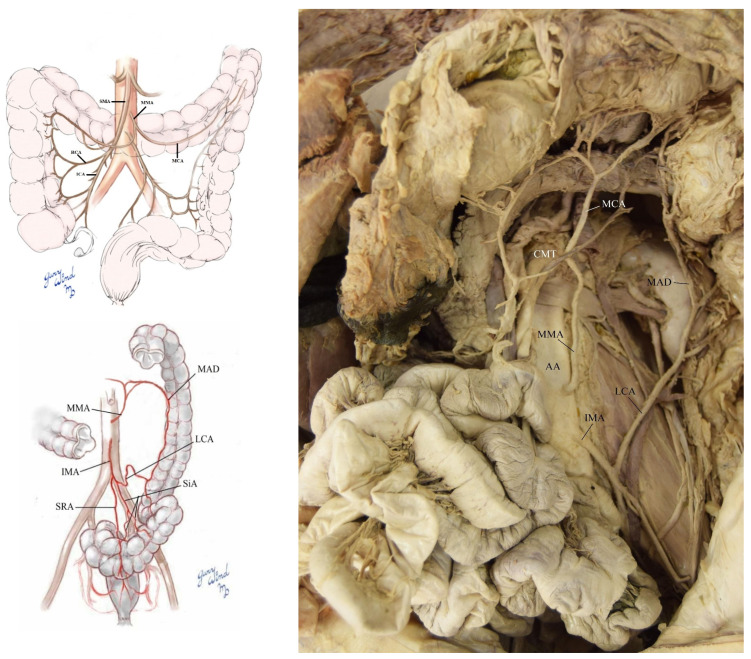
Schematic and gross image of the superior mesenteric artery (SMA), middle mesenteric artery (MMA) and inferior mesenteric artery (IMA) vasculature (clockwise from top left image).

## Data Availability

No new data were created or analyzed in this study. Data sharing is not applicable to this article.

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
