# Peer review of "A Hepatogastrophrenic Trunk, Celiacomesenteric Trunk, and a Middle Mesenteric Artery in a 68-Year-Old White Male Donor"

_diagnostics, 2022, doi:10.3390/diagnostics12071597_

Round 1

Reviewer 1 Report

In the manuscript entitled: A Hepatogastrophrenic Trunk, Celiacomesenteric Trunk, and a Middle Mesenteric Artery in a 68 year-old White Male Donor, the authors described a detailed analysis of a concordance of variations of the visceral branches of the abdominal aorta artery. It is an interesting paper that highlights the importance of anatomical variations and their clinical impact.

Although the authors cite Gielecki et al.,2005, I believe that for the manuscript to be complete, three important citations should be included in the bibliography: The chapter 54 of the Abdominal aorta of Daisy Sahni et al in Bergman's Comprehensive Encyclopedia of Human Anatomic Variation, 2016 and Michels NA. 1955. Blood Supply and Anatomy of the Upper Abdominal Organs with a Descriptive Atlas. Philadelphia, Montreal: JB Lippincott Company.

Michels NA. 1962. The anatomical variation of the arterial pancreatic,

bladder, bile duct, liver, pancreas and parts of small and large intestine.

J Int Surg 18:13–40.

Author Response

Thank you for your feedback on our submitted article entitled “A Hepatogastric Trunk, Celiacomesenteric Trunk, and a Middle Mesenteric Artery in a 68 Year-Old White Male Donor”. We have addressed all of your comments below and we appreciate you taking the time to review our edits.

Drs. Sheridan, Reynolds, Maynes, Wind, Leighton, and Granite

Although the authors cite Gielecki et al.,2005, I believe that for the manuscript to be complete, three

important citations should be included in the bibliography: The chapter 54 of the Abdominal aorta of Daisy Sahni et al in Bergman's Comprehensive Encyclopedia of Human Anatomic Variation, 2016 and Michels NA. 1955. Blood Supply and Anatomy of the Upper Abdominal Organs with a Descriptive Atlas. Philadelphia, Montreal: JB Lippincott Company. Michels NA. 1962. The anatomical variation of the arterial pancreatic, bladder, bile duct, liver, pancreas and parts of small and large intestine.

J Int Surg 18:13–40.

These citations have been added to the manuscript. Thank you very much for the suggestions!

Reviewer 2 Report

This manuscript describes a complex vascular variation of the abdomen, revealed during routine dissection, together with related clinical and embryological questions. Such complex combined variations have often been reported in the past, as a consequence of an embryonic “traffic jam”.

However, there are numerous points to be addressed and clarified.

Despite a long bibliography, some important monographs were omitted, e.g. VanDamme J-P, Bonte J. 1990. Vascular anatomy in abdominal surgery; Bergman RA, Thompson SA, Afifi AK, Saddeh FA. 1988. Compendium of human anatomic variation: text, atlas, and world literature.

Instead of repeating countlessly the legends for the abbreviations, a simple abbreviation list at the beginning of the manuscript would suffice.

The hepatogastrosplenic (HGS) trifurcation is also called the tripod according to Haller. The classification of the celiac trunk becomes easy if one consideres only the three main stems (SA, CHA and LGA); the other vessels are only smaller collaterals.

The case description lacks antecedents other than cause of death.

The legal background for this body donation must be addressed.

There is no morphometry, e.g. interarterial distances between origins of trunks (HGPT-CMT-MMA-IMA), or external diameters at base.

Figs 1 and 2a can be fused, or schema given as an inlay.

The right portion of Fig. 2a seems to be out of focus.

Fig. 2b is surplus.

There is no illustration nor description of the arborization of the SMA, is there a right colic artery, what is the course of the ileocolic artery, etc.

The MMA can also be interpreted as an anomalous origin of the middle colic artery (Yoshida et al. SRA 1993).

Figs 4 and 5 can be fused, or schema given as an inlay.

Figs 6 and 7 are surplus.

Pg 8, there is no need to describe in detail what is a normal finding. The definition of the spleno-mesenteric confluence is odd; it is conventional to say that the inferior mesenteric vein opens into the superior mesenteric vein, and that splenic and superior mesenteric vein constitute the portal vein. The bifid (double) gonadal vein is not such a rare finding, according to Lalawani (J Nat Sci Biol Med 2017) the incidence is 10%.

Pgs 9 and 10, the incidence of CMT is unclear in relation to the population (only “anomalies” of the CT, or whole population).

How was the ALHA defined as accessory? Was there an intrahepatic dissection of blood vessels?

The embryonic development should be corrected. The vitelline arteries are initially paired, and LGA, CHA and SA do not arise from separate roots, but from the 10th segmental vessel, and SMA from the 13th. The IMA arises from the 22nd vitelline artery. It would be very helpful if the authors consulted the Murakami typology of the celiaco-mesenteric arterial pattern (1995 and 1998).

The Appleby procedure was originally described as an extensive operation for advanced gastric carcinoma including linitis plastica.

The following: “...treatment of hepatocellular carcinoma. The right IPA classically provides vascular supply to this tumor” is an overstatement, extrahepatic collaterals are encountered in 27% of the cases (Kim et al., Radiographics 2005).

The phrase “Preoperative diagnostic scans with CT angiography may provide better visualization of vascular anomalies and purport surgical decision making” should be supported by a reference.

The Conclusion is too generalized. As for the statement “The frequency for these multiple variations to occur independently in one individual is estimated to be very low and has not increased despite improved imaging techniques”, is this just opinion of the authors, or are there any comparative study evidences?

Minor corrections:

Pg 2, 2nd paragraph: insert “also” between “variations” and “include”

Pg 7, capital “D” in “drummond”

Author Response

Thank you for your feedback on our submitted article entitled “A Hepatogastric Trunk, Celiacomesenteric Trunk, and a Middle Mesenteric Artery in a 68 Year-Old White Male Donor”. We have addressed all of your comments below and we appreciate you taking the time to review our edits.

Drs. Sheridan, Reynolds, Maynes, Wind, Leighton, and Granite

1) Despite a long bibliography, some important monographs were omitted, e.g. VanDamme J-P, Bonte J. 1990. Vascular anatomy in abdominal surgery; Bergman RA, Thompson SA, Afifi AK, Saddeh FA. 1988. Compendium of human anatomic variation: text, atlas, and world literature.

This citation, Sahni et al 2016, Bergman et al 1988, Michels 1955, Michels 1962, and Haller 1756 have been added to the manuscript. Thank you very much for the suggestion!

2) Instead of repeating countlessly the legends for the abbreviations, a simple abbreviation list at the beginning of the manuscript would suffice.

The legends have been revised.

3) The hepatogastrosplenic (HGS) trifurcation is also called the tripod according to Haller. The classification of the celiac trunk becomes easy if one considered only the three main stems (SA, CHA and LGA); the other vessels are only smaller collaterals.

We have added the reference to Haller into the manuscript.

4) The case description lacks antecedents other than cause of death.

The Maryland State Anatomy Board only provides the age, sex, ancestry, and cause of death of their donors.

5) The legal background for this body donation must be addressed.

This donor was one of fifty donors acquired from the Maryland State Anatomy Board for the Class of 2025 first year medical school anatomy courses. This information was added to the manuscript.

6) There is no morphometry, e.g. interarterial distances between origins of trunks (HGPT-CMT-MMA-IMA), or external diameters at base.

We have added morphometric measurements to the case description.

7) Figs 1 and 2a can be fused, or schema given as an inlay.

As a collective, we believe keeping the schematic as Figure 1 and the cadaveric image as Figure 2 would best represent the two figures, rather than inlaying or fusing them.

8) The right portion of Fig. 2a seems to be out of focus.

At the forefront of the image is the transverse colon and that is somewhat blurry in order for the camera to focus on the celiac trunk vasculature; the highlighted aspect of the image.

9) Fig. 2b is surplus.

Deleted Figure 2b.

10) There is no illustration nor description of the arborization of the SMA, is there a right colic artery, what is the course of the ileocolic artery, etc.

We have added a schematic of the SMA to Figure 4 with normal branching of the ileocolic and right colic arteries, as well as an anastomosing branch connecting with the MCA branching from the MMA.

11) The MMA can also be interpreted as an anomalous origin of the middle colic artery (Yoshida et al. SRA 1993).

Thank you for this additional information! It has been added under the discussion section 3.4.

12) Figs 4 and 5 can be fused, or schema given as an inlay.

Figures 4 and 5 are now one figure and listed as Figure 4. We have also added a SMA schematic to Figure 4.

13) Figs 6 and 7 are surplus.

Deleted figures 6 and 7.

14) Pg 8, there is no need to describe in detail what is a normal finding. The definition of the spleno-mesenteric confluence is odd; it is conventional to say that the inferior mesenteric vein opens into the superior mesenteric vein, and that splenic and superior mesenteric vein constitute the portal vein. The bifid (double) gonadal vein is not such a rare finding, according to Lalawani (J Nat Sci Biol Med 2017) the incidence is 10%.

We have deleted this section of the manuscript.

15) Pgs 9 and 10, the incidence of CMT is unclear in relation to the population (only “anomalies” of the CT, or whole population).

Defined study population as the 252 dissections of Japanese cadavers per reference article

16) How was the ALHA defined as accessory? Was there an intrahepatic dissection of blood vessels?

ALHA was defined as accessory because of its vascular course to the liver that mimicked the course of the LHA. The ALHA was an additional vessel following the same pathway as the LHA. We did not perform any intrahepatic dissections.

17) The embryonic development should be corrected. The vitelline arteries are initially paired, and LGA, CHA and SA do not arise from separate roots, but from the 10th segmental vessel, and SMA from the 13th. The IMA arises from the 22nd vitelline artery. It would be very helpful if the authors consulted the Murakami typology of the celiaco-mesenteric arterial pattern (1995 and 1998).

The articles have been reviewed and we believe the summary is adequate and correct.

18) The Appleby procedure was originally described as an extensive operation for advanced gastric carcinoma including linitis plastica.

Added this fact to the manuscript. Thank you for this information!

19) The following: “...treatment of hepatocellular carcinoma. The right IPA classically provides vascular supply to this tumor” is an overstatement, extrahepatic collaterals are encountered in 27% of the cases (Kim et al., Radiographics 2005).

This sentence has been updated in the manuscript.

20) The phrase “Preoperative diagnostic scans with CT angiography may provide better visualization of vascular anomalies and purport surgical decision making” should be supported by a reference.

This is more of a clinical conclusion—if anomalies provide difficulty during surgical operations, diagnostics scan for surgical planning may provide better visualization and support decision making. If a known anomaly exists and surgery has to take place, preoperative scans for planning is a logical conclusion. We had decided to change the sentence to say “of vascular anomalies in certain clinical cases and purport…”

21) The Conclusion is too generalized. As for the statement “The frequency for these multiple variations to occur independently in one individual is estimated to be very low and has not increased despite improved imaging techniques”, is this just opinion of the authors, or are there any comparative study evidences?

We aren’t quite sure what should be changed about the conclusion since the usual purpose of the conclusion section of a case study is to summarize the findings and how knowledge of such findings can be applied clinically. The article review that was compiled by the authors did not find any cases with these multiple anomalies, and for example (https://www.ncbi.nlm.nih.gov/pmc/articles/PMC3389861/) throughout our research, could not find any similar cases of patients with all these anomalies.

22) Pg 2, 2nd paragraph: insert “also” between “variations” and “include”

Added, thanks!

23) Pg 7, capital “D” in “drummond”

Corrected, thanks!

Round 2

Reviewer 2 Report

The manuscript is acceptable for publication, after the required comments have been considered and implemented.